# Root-Associated Bacteria Are Biocontrol Agents for Multiple Plant Pests

**DOI:** 10.3390/microorganisms10051053

**Published:** 2022-05-19

**Authors:** Jang Hoon Lee, Anne J. Anderson, Young Cheol Kim

**Affiliations:** 1Agricultural Solutions, BASF Korea Ltd., Seoul 04518, Korea; beau99@naver.com; 2Department of Biological Engineering, Utah State University, Logan, UT 84322, USA; annejanderson33@gmail.com; 3Department of Applied Biology, College of Agriculture and Life Sciences, Chonnam National University, Gwangju 61186, Korea

**Keywords:** biofilm, dual biocontrol, secondary metabolites, induced plant resistance, plant pathogens, insect pests, nematode pests, volatiles

## Abstract

Biological control is an important process for sustainable plant production, and this trait is found in many plant-associated microbes. This study reviews microbes that could be formulated into pesticides active against various microbial plant pathogens as well as damaging insects or nematodes. The focus is on the beneficial microbes that colonize the rhizosphere where, through various mechanisms, they promote healthy plant growth. Although these microbes have adapted to cohabit root tissues without causing disease, they are pathogenic to plant pathogens, including microbes, insects, and nematodes. The cocktail of metabolites released from the beneficial strains inhibits the growth of certain bacterial and fungal plant pathogens and participates in insect and nematode toxicity. There is a reinforcement of plant health through the systemic induction of defenses against pathogen attack and abiotic stress in the plant; metabolites in the beneficial microbial cocktail function in triggering the plant defenses. The review discusses a wide range of metabolites involved in plant protection through biocontrol in the rhizosphere. The focus is on the beneficial firmicutes and pseudomonads, because of the extensive studies with these isolates. The review evaluates how culture conditions can be optimized to provide formulations containing the preformed active metabolites for rapid control, with or without viable microbial cells as plant inocula, to boost plant productivity in field situations.

## 1. Introduction

Globally, there is a strong movement for sustainable and regenerative agriculture, where plant quality, yield, and soil health are key factors. Additionally, agricultural methods for disease control are under scrutiny by consumers. These factors increasingly favor biopesticides as an alternative to a strict synthetic chemical approach to pest control. The success of chemical pesticides reveals the high standards for efficacy that has to be matched by biopesticides. Chemical pesticides are formulated to consistently control plant diseases and pests at a relatively low cost to the user. The chemicals are often targeted toward one or a group of related pests. However, the use of these chemical pesticides poses risks such as environmental contamination with residues, development of resistance to the pesticide in the targets, and impacts on human health [1]. Regulation of the use of chemical pesticides is constantly changing, and it is challenging to develop and register new compounds [2]. Furthermore, because the industry has provided pesticides with specific targets for either microbial plant pathogens, insects, or nematodes, they are not broad-spectrum. Therefore, different applications, possibly at separate times, are required for complete plant protection. Multiple applications increase costs to the farmer. One recent exception is the synthetic pesticide fluopyram, commercialized as both a fungicide and nematicide, that has been well received by growers for its reliable performance [3]. Fluopyram is a succinate dehydrogenase inhibitor that targets various plant fungal pathogens and kills pathogenic nematodes [3,4]. However, there are nontarget effects at higher concentrations, including the induction of mammalian tumors [5].

Successful biopesticides will protect the plants while maintaining crop quality and yield, and nurture soil health while minimizing their hazardous effects on consumers and the soil environment. However, application of biopesticides in agricultural fields has drawbacks that research involved in formulation, storage, and applications must overcome [6]. In the field, the survival and efficacy of microbial inocula may be limited by environmental factors [7]. For example, agricultural chemicals may negatively influence the activity and survival of beneficial microbes in the field. The formulation of current chemical pesticides maximizes their efficacy through the development of a stable shelf life and optimal methods of field application. A long shelf life and stability under storage are problems for live cell formulations. Compared with the rapid killing effects of some pesticides, time is required to amass beneficial microbial communities in the rhizosphere. The early observations of biological control in the field of the take-all fungus, *Gaeumannomyces graminis*, on cereals demonstrate how time and repetitive cropping are required to establish a suppressive soil effective for this pathogen [8].

Laboratory studies have firmly established that certain plant-associated microbes afford biocontrol of bacterial and fungal plant pathogens as well as plant-consuming insects and nematodes. Table 1 lists some of the plant-associated microbes where the microbial metabolites active in biocontrol of multiple plant pests are characterized. Formulations of such biocontrol agents are attractive because they could simultaneously deter microbial disease and insect/nematode damage to plants. Although endophytic fungal entomopathogens also limit microbial disease pressure in plants, this review focuses only on biocontrol bacteria [9,10].

The review is structured into four information sections. Section 2 introduces major bacterial genera and species for which biocontrol of multiple species by characterized metabolites are established. The different classes of biocides characterized from these bacteria are introduced and reveal that some active metabolites are shared between genera. The findings illustrate widespread variability in the susceptibility of the targets, something that is true for both different plant fungal pathogens and insect predators. Section 3 focuses on how certain microbial metabolites with direct biocontrol activity also induce systemic tolerance in the plant to pathogens, damaging insects, or nematode pests. Section 4 addresses the integration of the academic knowledge of biocontrol with industrial expertise and mechanization for scale-up to formulate commercial preparations for field applications as biopesticides. The conclusions section summarizes the highlights of the review, showing the potential for biocontrol of multiple targets by rhizosphere microbes.

**Table 1 microorganisms-10-01053-t001:** Rhizosphere bacteria that protect plants against multiple pests (i.e., microbes, insects, and nematodes) through characterized metabolites and enzymes.

Bacterial Strains	Pathogens	Pests	Active Metabolites		Reference
			Antimicrobial	Insecticidal	
**Gram-positive bacteria**					
*Bacillus amyloliquefaciens* AG1	Various fungal pathogens	*Tuta absoluta*	Crude protein extract	Biosurfactant	[11,12]
*Bacillus atrophaeus* L193	*Botrytis cinerea*, *Monilinia laxa*	*Rhopalosiphum padi*	2,3-Butanediol	Biosurfactant	[13,14]
*Bacillus subtilis* PTB185	Various fungal pathogens	*Aulacorthum solani* *Aphis gossypii*	Lipopeptides	Chitinase	[15,16]
*Bacillus subtilis* SPB1	*Fusarium solani* *Rhizoctonia bataticola* *Rhizoctonia solani*	*Ectomyelois ceratoniae* *Spodoptera littoralis*	Biosurfactant	Biosurfactant	[17,18,19,20]
*Bacillus subtilis* V26	*Botrytis cinerea*	*Tuta absoluta*	Biosurfactant	Biosurfactant	[21]
*Bacillus thuringiensis* strains	*Sclerotinia sclerotiorum*	*Plutella xylostella*	Salicylic acid, ethylene, and jasmonic acid	BT toxin	[22]
*Bacillus thuringiensis* CMB26	*Colletotrichum gloeosporioides*	*Pieris rapae crucivora*	Lipopeptide	Lipopeptide	[23]
*Brevibacillus laterosporus* Lak1210	*Fusarium equiseti*	*Plutella xylostella*	Chitinase	Chitinase	[24]
*Paenibacillus elgii* HOA73	*Botrytis cinerea* *Cladosporium* *sphaerospermum*	*Plutella xylostella* *Meloidogyne incognita*	Chitinase	Crude enzyme, gelatinase, chitinase	[25,26,27]
*Paenibacillus elgii* HOA73	*Botrytis cinerea*, *Rhizoctonia solani*, *Fusarium oxysporum* f. sp. *lycopersici*		Methyl-2,3 dihydroxybenzoate (phenolic compound)		[28]
*Paenibacillus elgii* HOA73	*Botrytis cinerea*, *Rhizoctonia solani*		Protocatechuic acid		[29]
*Paenibacillus polymyxa* BMP-11	Various fungal pathogens	*Tribolium castaneum*	1-Octen-3-ol benzothiazole	1-Octen-3-ol benzothiazole	[30]
*Streptomyces hydrogenans* DH16	Various fungal pathogens	*Spodoptera litura* *Meloidogyne incognita*	NA	NA	[31,32]
*Streptomyces tanaschiensis*	*Saccharomyces* sp. *Penicillium* sp.	*Musca domestica* *Locusta migratoria*	Flavensomycin	Flavensomycin	[33]
**Gram-negative bacteria**					
*Photorhabdus temperata* M1021	*Phytophthora capsica*, *Rhizoctonia solani*, *Corynespora cassiicola*	*Galleria mellonella*	Benzaldehyde	Benzaldehyde	[34]
*Pseudomonas fluorescens* CHA0	*Pythium ultimum*	*Spodoptera littoralis* *Heliothis virescens* *Plutella xylostella* *Meloidogyne javanica* *M. incognita*	Pyoluteorin, 2,4-DAPG *	Pyoluteorin, 2,4-DAPG extracellular protease, and insect toxin (Fit **)	[35,36,37,38]
*Pseudomonas chlororaphis* PCL1391	*Fusarium oxysporum* f. sp. *radicis-lycopersici*	*Spodoptera littoralis* *Heliothis virescens* *Plutella xylostella* *Galleria mellonella*	Phenazine-1-carboxamide	Potent insect toxin, HCN, lipopeptide, Fit toxin	[38,39,40]
*Pseudomonas chlororaphis* O6	*Rhizoctonia solani* *Fusarium graminearum* *Phytophthora infestans*	*Meloidogyne hapla* *Myzus persicae*	Pyrrolnitrin, phenazines	Hydrogen cyanide (HCN), Cyclic lipopeptides	[41,42,43,44]
*Pseudomonas chlororaphis* PA23	*Sclerotinia sclerotiorum*	*Caenorhabditis elegans*	Pyrrolnitrin	Pyrrolnitrin, HCN	[45,46]
*Serratia entomophila* AB2	*Aspergillus flavus* *Candida albicans* *Fusarium oxysporum*	*Heliothis armigera*	NA	NA	[47]

* 2,4-DAPG: 2,4-diacetylphloroglucinol, ** Fit: for *P. fluorescens* insecticidal toxin.

Targeted virulence against plant pests and other beneficial effects that promote plant vigor fits the description of these bacteria as probiotics, a term applied to microbes that improve the growth of their hosts [48,49,50]. The rhizosphere habitat of the beneficial microbes enables the plant to be surrounded by overlapping spheres of influence that depend on different mechanisms for plant protection. This concept is illustrated in Figure 1. Central and throughout the plant is the induction of systemic resistance through root colonization by certain beneficials. The formation of biofilms of the beneficial microbes in part physically protects the root surface. Protection is extended further into the rhizosphere through the secretion of antagonistic enzymes or metabolites by the biofilm cells. The production of antagonistic volatiles provides an even greater radius of protection. 

## 2. The Array of Isolates and Their Products for Biocontrol of Multiple Targets

Bacteria with multiple biocontrol potential include genera classified as Gram-positive cells, such as the firmicutes *Bacillus*, *Paenibacillus*, and *Brevibacillus*; actinomycetes such as *Streptomycete* isolates; and Gram-negative isolates, including *Pseudomonas*, *Photorhabdus*, and *Serratia* (Table 1). For commercial formulation, Gram-positive isolates that sporulate are advantageous because the spores have an extended longevity over vegetative cells. Common habitats for these genera are soils. Many are documented to colonize root tissues, to which they are attracted by chemotaxis towards the gradient of plant root exudates. Certain isolates display very specific habitats, such as the symbiosis of *Photorhabdus luminescens* with entomopathogenic nematodes [51].

Biocontrol-active metabolites are diverse (Figure 2) but can be classified based on their structural similarity; for example, peptide toxins are implicated in insect and nematode control. Indeed, the most commercially relevant are the toxins produced by *Bacillus thuringiensis*, collectively termed BT toxins, which function by generating holes in the membranes of the insect’s digestive tract [52]. Fit proteins from the *fit* genes in the genomes of certain pseudomonads [53] and the related Mcf toxins from *Photorhabdus* and *Xenorhabdus* [54] that induce membrane disorganization also contribute to insecticidal activity. The lipopeptide group impacts membrane structures through their surfactant activity [55]. Phenolics such as the phenazine group may cause oxidative stress in the target leading to cell death, and some may act as iron chelators [56]. The hydrolytic enzymes chitinase, proteases, and lipases could damage cellular structures [37,57,58]. The combined activities of proteases and lipases could impair membrane structure. The chitinases could hydrolyze cell wall polymers in insects, nematode egg coats, and fungi [58]. 

The direct effects, revealed by dose-dependent in vitro studies with purified materials, generally require high active metabolite or enzyme levels. However, in addition to the potential direct effects of the above metabolites and secreted enzymes, there is also a strong involvement by the same active biocontrol product in activating systemic protection measures in the plant [59]. Characterized traits of the microbes and functions of the metabolites with potential biocontrol significance are summarized by type in Figure 2. The following discussion introduces the rhizosphere habitat for biocontrol agents and, then, with a focus on firmicutes and pseudomonads, the biocontrol potential of characterized metabolites produced by different isolates are compared. 

### 2.1. The Rhizosphere Habitat: A Significant Trait of Microbes with Multiple Biocontrol Activities

The rhizosphere is a dynamic space and key to the health and productivity of the plant. Roots and their exudates supply nutrients to damaging microbial plant pathogens and a foothold for invasion by pathogenic nematodes and insect larvae. However, the plant counters these effects by nurturing root colonization by beneficial microbes that shift the balance towards root and shoot health. Firmicutes and Gamma-proteobacteria, such as the pseudomonads, are among the most-studied bacteria that are such probiotics for the plant [60]. Their consumption of simple metabolites, such as the sugars and amino acids in root exudates [60,61], reduces the ability of root exudation to support the growth of pathogenic microbes, insect larvae, and nematodes. Further, the microbe’s metabolism increases the level of protectant secondary microbial products in the soil pore waters. These protective metabolites may require specific catabolic mechanisms in order to have persistence and to influence microbial community composition [62]. One example by Stringlis et al. [63] shows how coumarins in the root exudates favor root colonization by biocontrol-active microbes with a tolerance to coumarin, whereas this root metabolite is toxic to pathogen growth. The pools of biocontrol-active metabolites will function to strengthen the niche of the beneficial microbes, and additionally may trigger systemically protective changes in the plant to further thwart potential pathogens.

The formation of biofilms by the probiotic cells on root cell surfaces is integral to the protection process [64]. The biofilm patches shield the root surface cells against direct attack and provide a sheltered environment for the cells of the beneficial bacterium. Additionally, the extracellular polymers forming the gel for the matrix encapsulating the cells could retain released microbial products, so that their concentrations are higher than in the soil pore water. When insect larvae and nematodes feed on the colonized plant root, they will ingest these biofilms that are loaded with toxic materials. Interestingly, both resorcinols and phenazines, identified as phenolics active in biocontrol, are correlated with improved biofilm formation [45,64,65]. The biofilm would also reduce the ingress of pathogenesis factors such as enzymes and toxins from any plant pathogens, so that contact with root cells is lessened.

Studies with a *Paenibacillus polymyxa* strain confirm that the matrix exopolysaccharides of the biofilm act as a rhizosphere nutrient source and as a structure that inhibits pathogen attack [66]. Chan et al. [67] indicate that reduced motility of nematodes within a *Pseudomonas aeruginosa* biofilm impairs their predatory behavior. Earlier work with the biocontrol agent *Pseudomonas brassicacearum* DF41 observed that biofilm formation over the head of the nematode contributes to the nematode’s demise [68]. This group cites the activity as being part of the mechanism that limits predation of the beneficial bacterium, i.e., it is part of the survival features of the biocontrol agent.

Invasion and feeding on root cells are part of the life cycle of the nematodes and insect larvae that damage plants. In this context, being able to kill these organisms would provide nutrients for the biocontrol organisms, thus augmenting the carbon-rich metabolites in the root exudates. A comparison of the root exudate composition shows that colonization of the roots by beneficial microbes alters the composition of amino acids [69,70,71]. Some amino acids presumably supply essential N for microbial growth, which is limited in the soil pore water. For instance, Boiteau et al. [70] found that the concentration of two N-containing amino acids in root exudates from *Brachypodium* decreases after colonization by *Pseudomonas fluorescens*. Microbes that are rhizosphere colonists have higher numbers of amino acid and sugar transporters than non-colonists [69]. Consequently, the presence of genes enabling the biocontrol agents to kill protein-rich larvae would be advantageous for bacterial multiplication, because of the rich supply of complex N-nutrients from the insect’s body mass [72,73]. Similarly, enzymatic digestion through proteases of the proteinaceous cuticle of nematodes or the chitin of their eggs would increase the N supply to the bacteria [74]. 

Infections with nematodes or insect larvae would result in biofilms on these hosts and the production of the biocontrol-active metabolites and enzymes, providing “hot spot” reservoirs additional to those at the plant root. Indeed, infection with insect larvae by biocontrol-active pseudomonads induces the expression of genes supporting antifungal metabolite production, boosting the value of such pockets of biocontrol microbes within the soil [40]. Further, any movement by the infected larvae and nematodes would aid in spreading the biocontrol agent (Figure 1). The concept that biocontrol microbes in the rhizosphere are spread by contact with and pathogenicity on insects is discussed well in a review by Pronk et al. [75]. Thus, the infectious ability on the soil fauna boosts the sphere of influence of the probiotics further from the root surface [60] (Figure 1). These potential roles by the biocontrol agents of the biofilm in plant protection add to the need to formulate viable cell preparations that are effective when introduced into agricultural settings.

### 2.2. Active Products from Bacillus spp.

Four classes of protectants essential in biocontrol by *Bacillus* spp. are peptide toxins, lipopeptides, enzymes, and volatile organic compounds. The most effective of these products are the insecticidal BT toxins, which have been successfully commercialized. The toxins are produced as sporulation of the *Bacillus* isolates commences and become concentrated as an inactive crystal structure within bacterial cells [52]. After ingestion of the crystals by the insect larvae, conversion to an active peptide dependent on the action of a larval protease occurs. The recognition of the active peptide by specific receptors on insect gut cells causes pore formation and loss of selective permeability. The efficacy of digestion is impaired, and the bacteria enter the circulatory systems to eventually kill the larvae. The specificity of the receptors in the gut cell membranes leads to products that target certain insects. The toxins vary in structure and activity; BT from *B. thuringiensis* shows an LC50 of 0.28 ppm against the second instar larvae of the diamondback moth *Plutella xylostella*, a value lower than the LC50 for the toxin from *Bt* subsp. *kurstaki* HD1 (LC50 0.47 ppm) [22]. However, BT toxins are not the only control mode, because additional toxic peptides can be formed even in vegetative cells [76]. Understanding the additive or synergistic effects of the array of metabolites in biocontrol requires further studies.

Lipopeptides are metabolites commonly secreted by *B. thuringiensis* and other *Bacillus* species [77]. Many lipopeptides have potent activity against insects and are also antifungal (Table 2). For example, the lipopeptide from *B. thuringiensis* CMB26 has insecticidal activity on the larvae of the cabbage white butterfly, *Pieris rapae crucivora*, and inhibits the growth of the pathogenic fungus, *Colletotrichum gloeosporioides* [23]. A single *Bacillus* spp. strain may produce several lipopeptides, each with a different potential against plant fungal pathogens. Two lipopeptides produced by *B. subtilis* EA-CB0015, iturin A and fengycin C, inhibit the anthracnose pathogen *Colletotrichum acutatum*, with a minimum inhibitory concentration (MIC) at 32 ppm and 124 ppm, respectively [78]. Other *Bacillus* species also produce iturin A, and this biocide also disrupts the membranes in cells of the fungal targets [79]. Studies of iturin from *B. methylotrophicus* TEB1 show a MIC of 100 ppm against *Phoma tracheiphila* [80], whereas iturin from *B. a**myloliquefaciens* MG3 inhibits mycelial growth of *C. gloeosporioides* at concentrations less than 50 ppm but is ineffective in restricting spore germination [79]. These examples indicate that timing is crucial for fungal pathogen control on the plant, as it will depend on whether the spores or mycelial growth of a pathogen is targeted. Another lipopeptide from *B. amyloliquefaciens* BO5A inhibits mycelia of *F. oxysporum* at 10 ppm although there is no effect on the mycelium of *Botrytis cinerea* at 100 ppm [81]. 

Regarding insecticidal activity, a lipopeptide from *B. amyloliquefaciens* AG1 has an LC50 of 180 ng/cm^2^ against larvae of the leaf miner, *Tuta absoluta*, targeting the membranes in the larval midgut cells [12]. These same larvae are controlled at 278 ng/cm^2^ by the lipopeptide from *B. subtilis* V26, which is also an antifungal that halts mycelial growth of the gray mold fungus, *B*. *cinerea*, at 2 ppm [21].

The surfactin lipopolypeptide from *B. subtilis* Y9 shows aphicidal activity against the green peach aphid *Myzus persicae* at 20 ppm [82]. Similarly, active surfactant lipopeptides from *Bacillus atrophaeus* L193 cause the destruction of the cuticle of the aphid *Rhopalosiphum padi* [13]. *B. atrophaeus* L193 also controls fungal diseases caused by *B. cinerea* and *Monilinia laxa* in cherry fruits by producing 2,3,-butanediol, a weakly volatile metabolite [14]. Butanediol production is also observed in other plant-associated biocontrol bacteria [92], again showing the sharing of beneficial traits.

Other dual biocontrol lipopeptides synthesized by *B. subtilis* SPB1 kill larvae of the cotton leafworm *Spodoptera littoralis*, initially through the disruption of midgut function [17]. The lipopeptide also effectively controls two fungal pathogens of potato and tomato, *Fusarium solani* and *Rhizoctonia solani*, but at high MICs of 3000 ppm and 4000 ppm, respectively, compared with an MIC of 40 ppm for *Rhizoctonia bataticola* [20]. Of note is the finding that the *B. subtilis* SPB1 lipopeptides have preventative and curative effects [19]. These findings illustrate the strong potential for lipopeptides to combat plant loss due to insects and fungal pathogens, however, the sensitivities between targets are different.

Although the lipopeptide from *B. subtilis* PTB185 effectively controls the gray mold fungus *B. cinerea*, its chitinase is the major factor combating the aphids *Aulacorthum solani* and *Aphis gossypii* [15,16]. Chitinase of *B. firmus* also is highlighted as a seed treatment agent to control nematodes [93]. Several papers discuss the multiple roles of chitinase regarding its antifungal, insecticidal, and nematicidal activities [58,94]. However, chitinase is not implicated in the control of the nematode *Meloidogyne incognita* by a crude supernatant from *B*. *firmus* YBf-10. The efficacy in pot cultures is comparable to that of the chemical nematicide fenamiphos [95]. In this case, a serine protease and chemical metabolites seem to be functional in the nematicidal response [96].

*Bacillus* strains also produce biocontrol-active volatile organic compounds (VOCs). Two VOCs generated by *B. velezensis* CT32, 2,4-dimethyl-6-tert-butylphenol and benzothiazole, have strong antifungal activity against wilt in strawberries caused by *Verticillium dahliae* and *F*. *oxysporum* [97]. These VOCs are documented in the headspace when the bacillus is grown on nutrient agar. Unidentified VOCs produced by another *B. velezensis* strain, isolate VN10, inhibit mycelial growth of *Sclerotinia sclerotium* with high efficacy (MIC 1 ppm) and reduce the disease caused by this fungus through suppression of its production of oxalic acid [98].

### 2.3. Paenibacillus spp. Metabolites and Enzymes

The biocontrol arsenal of *Paenibacillus elgii* HOA73 displays the common finding that a multiplicity of metabolites formed by a single isolate is important in biocontrol. Hydrolytic enzymes (chitinase, protease, and gelatinase), along with the phenolic compounds benzothiazole, methyl 2,3-dihydroxybenzoate, protocatechuic acid, and the volatile 1-octen-3-ol, are among its biocontrol weapons. The phenolic compounds display promising MIC values for fungal pathogen control. Methyl 2,3-dihydroxybenzoate inhibits mycelial growth of *B. cinerea*, *R*. *solani*, and *F. oxysporum* f. sp. *lycopersici* at 32–64 ppm [28], and the MIC for protocatechuic acid is similar for *B. cinerea* and *R*. *solani* at 64 ppm. At 100 ppm, protocatechuic acid halts the disease progress of gray mold on strawberry fruits [29]. The lipopeptides from *Paenibacillus* strains also participate in biocontrol. A lipopeptide from *P. polymyxa* shows antimicrobial and anti-insect efficacy [66].

Active enzymes include chitinase, which inhibits spore germination of *Cladosporium tenuissimum* and *B*. *cinerea* at 100 ppm, whereas spores of *Fulvia fulva* and *C. gloeosporioides* are resilient [99]. The *P. elgii* HOA73 strain exhibits insecticidal and nematicidal performance correlating with chitinase and gelatinase (a type of protease) activities; mortality of *M*. *incognita* J2 juveniles and a reduction in egg mass and galling in tomatoes occur at 50–400 ppm. Other factors may participate in the nematicidal effects and the killing of second instar larvae of the diamondback moth, *P. xylostella*. Combining organic sulfur with the active enzymes acts synergistically [26]. It will be interesting to see how biocontrol microbes and their products can be combined with other tools to achieve better control.

VOCs from *P. polymyxa* BMP-11 are implicated in biocontrol. Two VOCs, 1-octen-3-ol (LC50, 16.75 ppm) and benzothiazole (LC50, 3.5 ppm), are active against the adult red flour beetle *Tribolium castaneum* [30]. In vitro work with authentic 1-octen-3-ol shows potent inhibition of bacteria and fungi at 1–2 ppm [100]. At 100 ppm, 1-octen-3-ol effectively inhibits mycelial growth of the brown rot fungus, *Monilinia fructicola*, but fruit treatment requires about 50 ppm to slow disease progress. Fumigation with 1-octen-3-ol (LD50, 27.7 mL/L air) reduces the effects of the maize weevil *Sitophilus zeamais* and the production of mycotoxin by the fungus *Fusarium verticillioides* (MIC 81.5 mL/L air) [101]. Citronellol, also produced by *P*. *polymyxa* strain BMP-11, inhibits *F. oxysporum.* Additionally, VOCs from *P*. *polymyxa* KM2501-1 can kill second-stage juveniles of *M*. *incognita* through mechanisms that include acting as a repellent and a fumigant active on contact [102].

These examples suggest that certain VOCs can be applied as fumigants. However, boosting the plant-associated bacteria in agricultural soils by inoculation would also promote beneficial volatile production in the rhizosphere. These methods could be valuable when raising crops in enclosed spaces, such as in greenhouse cultivation.

### 2.4. Pseudomonad Products

The weaponry of biocontrol pseudomonads uses the same metabolite classes discussed for the firmicutes in Section 2.2 and Section 2.3 for multiple control activity of nematodes, insect larvae, and microbial plant pathogens. Toxic peptides, lipopeptides, and volatiles are significant, as are several classes of phenolic-based structures (Figure 2).

Among the phenolic-based products with biocontrol activity, phenazines are well-known for their potential as antifungal compounds [103,104]. In general, phenazine-1-carboxylic-acid (PCA) has higher antifungal activity than phenazine-1-carboxamide (PCN) or 1-hydroxyphenazine (1HP) [105,106]. Phenazines of different structures can be produced simultaneously by pseudomonads, dependent on their gene pools [107].

The commercial product Shenqinmycin contains PCA as the major metabolite and has high efficacy in the field against *Phoma* infections [108]. The PCA produced by *P*. *fluorescens* 2–79 limits the mycelial growth of several plant pathogens; *Cochliobolus sativus*, *G. graminis* var. *tritici*, and *R*. *solani* are inhibited with MICs of 1 ppm, but *Fusarium* spp. requires 25–30 ppm. Species-dependent sensitivity is seen in the oomycete *Pythium*, with MIC values for PCA of 1 ppm for *Pythium aristosporum*, 25–30 ppm for *Pythium ultimate*, and 80–100 ppm for *Pythium ultimum* var. *sporangiifurum* [87].

Although mycelial exopolysaccharide production from *B. cinerea* is inhibited at 3 ppm PCA, the control of postharvest gray mold caused by this fungus requires 25 ppm [83]. The PCA synthesized by *P*. *aeruginosa* GC-B26 is active against *Colletotrichum orbiculare* as well as the oomycetes *Pythium capsici* (MIC 5 ppm) and *P*. *ultimum* (MIC 5 ppm) [84]. Studies with PCA from *P. aeruginosa* report activities against *Sclerotium rolfsii* (MIC 29 ppm), *F*. *oxysporum* (MIC 40 ppm), and *Colletotrichum falcatum* (MIC 50 ppm) [85]. The effective dose of PCA is often similar to the doses required for commercial pesticides, for example, for protection against Phytophthora blight on pepper and anthracnose on cucumber [84].

The control of tomato foot and root rot, caused by *F. oxysporum* sp. *radices-lycopersici*, with *P. chlororaphis* PCL1391 uses PCN as the major phenazine [39]. As for PCA, effective doses of PCN are similar to those of chemical products. Extracted PCN, produced by *P*. *aeruginosa* MML2212, effectively controls rice sheath blight caused by *R*. *solani* at 5 ppm, exceeding protection from the chemical fungicide carbendazim. Control of rice bacterial leaf blight, *Xanthomonas oryzae* pv. *oryzae*, by 5 ppm PCN is similar to that by the pesticide refamycin [89], and control of *B. cinerea* by PCN from *P*. *aeruginosa* at 108 ppm has almost the same inhibition rate as the chemical fungicide carbendazim. The effective dosages for PCN and chemical fungicides are higher for effects on spore germination and mycelial growth of *Sclerotinia*, requiring about 700 ppm for both PCN and carbendazim [88].

In contrast to their high efficacy for inhibiting various plant fungal pathogens, phenazines are not as efficient in suppressing insect or nematode larvae. They may impair egg hatching and promote increased J2 larval mortality of the root-knot nematode *M*. *incognita* [109,110]. Among the secondary metabolites produced by *P. chlororaphis* strain PA23, hydrogen cyanide (HCN) and the phenolic-based pyrrolnitrin (PRN) are cited as active agents in killing the nematode *Caenorhabditis elegans*, for which they also act as repellents. Growing *P. chlororaphis* strain PA23 with a nematode as the food source enhances the production of HCN and PRN [46]. However, mutational analysis of *P*. *chlororaphis* PA23 reveals that, among the secondary metabolites produced by this bacterium, pyrrolnitrin is the major metabolite involved in inhibiting *S*. *sclerotiorum* [45]. Studies with root-knot nematode juveniles found that HCN production is essential for the nematicidal effects of *P. chlororaphis* O6 [41,43].

Induced mortality of the root-knot nematode by *P. fluorescens* CHA0 requires another suite of products involving a protease and two phenolic-based structures, pyoluteorin and 2,4-diacetylphloroglucinol (2,4-DAPG), each with demonstrated antifungal activities [36,37]. The production of 2,4-DAPG by *P. fluorescens* CHA0 also suppresses populations of another nematode, *Meloidogyne javanica* [36], while 2,4-DAPG from *P. fluorescens* is toxic to *Xiphinema americanum* adults with an LC50 of 8.3 ppm. Apparently, 2,4-DAPG does not harm beneficial entomopathogenic nematodes [91].

The antifungal activity of 2,4-DAPG from pseudomonad isolates is well documented. It is effective at 50 ppm against the citrus postharvest fungi *Penicillium digitatum* and *Penicillium italicum*, a dose that compares favorably to 600 ppm for the chemical fungicide albesilate [111]. Production of 2,4-DAPG by *P. fluorescens* CHA0 is a major factor in suppressing the soil-borne disease black root rot in tobacco, caused by *Thielaviopsis basicola*, and against *G. graminis* var. *tritici* in wheat [90].

Other metabolites play roles in the insecticidal activities displayed by the multi-biocontrol pseudomonads [38,53]. Flury et al. [40] proposed that a mixture of the cyclic lipopeptide orfamide A and volatile HCN contributes to the insecticidal activity of *P. fluorescens* CHA0 and other pseudomonads [40]. Interestingly, the growth of *P. fluorescens* CHA0 on insect larvae promotes the expression of genes involved in producing an array of antifungal metabolites from this strain [40]. This finding suggests the to this rich nutrition of the bacterium helps its arsenal of metabolites provide security against competition with invasive microbes in the rhizosphere and bulk soil. Even though HCN is produced only at low levels by *P. protegens* Pf-5, the synthesis of analogs of rhizoxin adds to the power of orfamide A and a chitinase in the oral toxicity of this isolate to the larvae of the common fruit fly, *Drosophila melanogaster* [112].

Proteins termed Fit toxins are another part of the pathogenic mechanism of *P. protegens* and *P. chlororaphis* isolates on lepidopteran larvae [38,53,113]. Ruffner et al. [54] discussed how the *fit* gene clusters in the pseudomonads are related to those in two insect pathogens, *Photorhabdus* and *Xenorhabdus*, suggesting horizontal gene transfer of the clusters into limited pseudomonad genomes.

Hydrogen cyanide, HCN, is an important biocontrol volatile from the pseudomonads and other rhizobacteria including *Bacillus* isolates [114,115,116], contributing to a broad spectrum of inhibitory activities against microbes, insects, and nematodes [40,117]. Root exudates may exogenously supply the glycine that the microbes use as the substrate for HCN synthase; carbon dioxide is the other product of the synthase activity [117]. Production of HCN is observed in the airspace of closed growth boxes supporting plants with roots colonized by *P. chlororaphis* O6 [43,117]. Dependent on concentration, HCN may be toxic due to the disruption of cellular function through the inhibition of heme group function [118]. Indeed, root colonization and HCN production by certain isolates leads to herbicidal activity [119]. However, lower doses of HCN may also be important through synergistic interaction with other antimicrobial metabolites, such as lipopeptides or pyrrolnitrin, in enhancing the biocontrol effects.

The inhibition of fungal growth of the foliar plant pathogens *Septoria tritici* and *Puccinia recondita* f. sp. *tritici* on wheat is attributed to HCN production by *P. putida* BK8661 [120]. Furthermore, cyanide-producing pseudomonads suppress fungal diseases on canola and rice [121,122]. Mutation of *P. fluorescens* CHA0 revealed that HCN is a major player in controlling black root rot of tobacco, caused by the fungus *Thielaviopsis basicola* [123,124]. The combination of HCN and 2,4-DAPG syntheses by *Pseudomonas* sp. LBUM300 protects tomatoes against bacterial canker caused by *Clavibacter michiganensis* subsp. *michiganensis* [125].

Insecticidal effects of HCN production have been revealed by the observed mortality of second instar nymphs of the green peach aphid *Myzus persicae* when exposed to volatiles from *P. chlororaphis* O6 [44]. This study, and others with different pseudomonad isolates, showed that HCN production is controlled by quorum sensing [44]. Other studies extend the insecticidal effects of HCN produced by pseudomonads to flies and termites [126,127,128]. Combining HCN with lipopeptides of *P. protegens* CHA0 and *P. chlororaphis* PCL1391 shows activity against insects [40]. The nematicidal activities of HCN include killing of the nematode *Caenorhabditis elegans* by *P*. *chlororaphis* PA23, in combination with pyrrolnitrin [122]. A primary role for HCN is implicated in the mortality of J2 juveniles of the root-knot nematode, where for tomato, the efficacy of the biocontrol agent was as strong as that of the chemical nematicide fosthiazate in commercial greenhouses [41,43].

These studies illustrate that the biocontrol-active pseudomonads, like the firmicutes, strongly influence rhizosphere health, their strength being dependent upon a multiplicity of potential control mechanisms incited by different metabolites. 

### 2.5. Streptomyces, Brevibacillus, Serratia, and Photorhabdus

As discussed above, secondary products from the firmicutes and pseudomonads produce many different structures with multiple biocontrol activities. Only a few highlights are provided in this section to illustrate the expansive diversity in structure and function. The Gram-positive *Streptomyces hydrogenans* DH16 secretes metabolites that kill *M*. *incognita* J2 juveniles at 100 ppm [32], inhibit fungal pathogens, and induce mortality to second instar larvae of *Spodoptera litura* [32]. A chitinase produced by *Brevibacillus laterosporus* Lak1210 has dual performance against larvae of the diamondback moth *P. xylostella* and the pathogenic fungi *Fusarium equiseti* [24].

For other Gram-negative isolates, the volatile benzaldehyde produced by *Photorhabdus temperata* M1021 contributes to the antimicrobial effects on *Phytophthora capsici*, *R. solani*, *Corynespora cassiicola*, and *Bacillus* spp., as well as insecticidal activity on the larvae of the greater wax moth *Galleria mellonella* [34].

*Serratia* species are a source of many deterrents for plant protection and represent an understudied resource for soil and plant associations [129]. Ordentlich et al. [130] reported that chitinase from *Serratia marcescens* is key for controlling *Sclerotium rolfsii*; direct lysis of mycelia of the pathogen was observed. *Serratia entomophila* AB2 shows a dual effect on pathogenic fungi and the pod borer *Heliothis armigera* [47]. Haterumalides, novel and complex metabolites from *Serratia plymuthica* A153, suppress the apothecial formation in sclerotia of *Sclerotium* with MIC of 0.5 ppm; additionally, oomycetes also are sensitive to these metabolites [131,132]. *Serratia* isolates are amongst the several bacterial genera that synthesize pyrrolnitrin, previously discussed as a pseudomonad product in Section 2. In combination, pyrrolnitrin and the haterumalides inhibit spore germination of various fungal plant pathogens at MICs of 0.4–50 ppm [131]. The array of volatiles synthesized from *Serratia* isolates also offers biological control potential [133]. Recent work on the plant-growth-promoting isolate of *S. plymuthica* A153 correlates impaired growth of several plant fungal pathogens with the volatiles for which ammonia is identified as an important component [133]. Activation of the WRKY18-plant stress pathway was noted with volatiles of *Serratia* [134].

These examples illustrate that the soil contains a diversity of microbial genera that can contribute to plant health by affecting the growth and metabolism of potentially harmful microbe, nematode, and insect challenges. The chosen examples highlight that many soil isolates thrive in the rhizosphere partly due to the carbon and nitrogen nutrition from plant metabolites in the root exudates. The examples, summarized in Figure 2, reveal a wide range of microbial metabolites are involved in the intricate interactions with cohabitors, and some directly reduce the effects of the plant predators. The sources of these metabolites, the biocontrol rhizobacteria, become the predators of the plant pests but retain balanced cohabitation with the plant.

## 3. Induction of Plant Systemic Resistance Mechanism by Biocontrol Agents

The observations of inhibited growth of fungal pathogens or the death of infested insect larvae or nematode juveniles established that plant-associated microbes produce compounds that directly inhibit pathogenic challenges. However, there is another valuable hidden trait: many microbes trigger the induction of systemic resistance mechanisms in the plant (Figure 1). The ability of beneficial microbes to induce resistance to both microbial plant pathogens and herbivorous insects/nematodes is summarized in several reviews [135,136].

In addition to causing direct pest antagonism, the microbial stimulation of plant tolerance also contributes to plant health. The onset of systemic plant resistance can be traced to the effects of discrete metabolites from the biological agents, indicating dual roles of these compounds as direct inhibitors of plant pests. This additional significance of biological control microbes to plant biotic resistance is discussed in Section 3.

Certain antifungal products from biocontrol-active pseudomonads and *Bacillus* isolates trigger plant resistance, as determined by treatment with the authentic compound. Additionally, this ability is detected from a comparison of responses that are induced in the plant by the wild type, but not by mutant cells defective in the metabolite. Resistance in the plant is seen at a distance from the site of microbial colonization; for example, although the stimulus is from a root colonizer, leaf tissues are protected against the pathogens. Thus, the term induced systemic resistance (ISR) is used for this process. 

A cross-talking set of pathways involving different plant regulatory signals is involved. Identified pathways included the salicylic acid pathway, the jasmonate pathway, and an ethylene pathway [137]. Each pathway has a set of defense measures associated with the specific signal, such that one pathway offers protection against different sets of challenges [135]. For example, jasmonic acid pathway activation is associated with the plant’s responses to chewing insects [138], whereas plant cell necrosis by toxins from a plant pathogen triggers the salicylic acid pathway [139].

Studies with *Pseudomonas* sp. CMR12a find that mutants lacking phenazine production fail to induce systemic resistance in rice, whereas it is induced by the wild type cells and the phenazine metabolite, PCN [140]. Another phenazine, pyocyanin, produced by *P. aeruginosa* 7NSK2, stimulates ISR in tomatoes [141]. In tobacco, mutants of *P. chlororaphis* O6 lacking in phenazines are unable to induce systemic resistance to the soft-rot-causing pathogen *Erwinia carotovora* [142]. The ethylene defense pathway, rather than the salicylic acid pathway, is induced by *P. chlororaphis* O6 [143]. One common response to phenazines in plants is the production of reactive oxygen species, which may be an initiator for plant defense gene expression changes leading to ISR.

With *P. chlororaphis* PA23, the production of pyrrolnitrin is a possible activator for low-level systemic resistance to blackleg in canola [144]. Different compounds, an alkaloid, two amino lipids, three arylalkylamines, and a terpenoid, are active ISR activators from another root colonizer, *P. fluorescens* N 21.4 [145]. These were found by screening the metabolite pool from this bacterium. The release of 2,4-DAPG from other pseudomonads activates the jasmonic acid/ethylene pathway for ISR in *Arabidopsis* [146]. Resistance induction by 2,4-DAPG correlates with the suppression of take-all in soils [147]. Additionally, the suppression of root-knot nematode infections in tomato roots is linked to induced resistance by 2,4-DAPG produced from *P. fluorescens* CHA0 [36].

Volatiles produced by biocontrol bacteria also trigger resistance [148]. Kishimoto et al. [149] find that authentic 1-octen-3-ol induces systemic resistance through the jasmonate pathway. This VOC is produced by several biocontrol microbes, as discussed in Section 2. Ryu et al. [150] identified 2,3-butanediol among microbial volatiles from *Bacillus subtilis* GB03 and *Bacillus amyloliquefaciens* N937 that induce systemic resistance to gray mold, and *B. cinerea* in *Arabidopsis*, where the ethylene pathway, but not the salicylic acid or jasmonate pathway, is active. Butanediol production from *P. chlororaphis* O6 also explains the induction of systemic resistance to *Erwinia* in tobacco [92,151].

The lipopeptides produced by many *Bacillus* species also trigger ISR. Stimulation of ISR by iturin A and fengycin from indigenous *Bacillus* spp. offers protection in rice. This mechanism adds to the direct inhibition of spore germination and mycelial growth to achieve a high level of plant protection [152]. Park et al. [153] show that several analogs of iturin A from *Bacillus vallismortis* strain EXTN-1 activate ISR against *Phytophthora capsici* in chili pepper. Indeed, soft rot disease in cherry tomato caused by *Rhizopus stolonifera* is controlled by ISR activated by iturin A [154]. In both iturin A studies, an array of plant genes encoding enzymes associated with plant resistance is expressed.

The chitinases secreted by many of the biocontrol agents may also participate in enhanced plant resistance. Increased expression of plant chitinase genes is part of the changes in metabolism seen when ISR is activated in the plant; the induced proteins are termed the “pathogenesis-related proteins”. Differences in the array of chitinase isozymes expressed are dependent on the activation of the salicylic or ethylene/jasmonic acid pathways [155,156]. These hydrolytic enzymes degrade chitins/chitosans that are integral components of the walls of fungi and insects and the coating of nematode eggs [157,158]. However, chitin oligomers derived from such activities are among the group of activators for innate immune responses in plants [156]. Consequently, the secretion by the bioactive bacteria of chitinases, often a complex of different isoforms [25], is yet another avenue by which these microbes can indirectly protect the plant through their potential to activate plant defenses.

## 4. Secondary Metabolites and Cell Preparations for Commercial Biocontrol Formulations

Several biopesticides based on biocontrol metabolites with a control potential as effective as chemical pesticides are currently marketed for agricultural use. One such example is fludioxonil, a synthetic fungicide derived by using pyrrolnitrin as a base structure. Therefore, it belongs to the phenylpyrroles group of pesticides (FRAC code 12). Fludioxonil inhibits signal transduction in fungi and effectively controls *Sclerotinia* and *Botrytis* in various crops. Thirty years after its development, there have been few reports of resistance to fludioxonil [159]. Another commercial product is shenqinmycin, based on PCA. It has been in use since 2011 to control fungal diseases [160]. Metabolites from *Streptomyces* spp. are the foundation for many other chemical pesticides. Insecticides and nematicides that mimic the natural products from *Streptomyces* spp. include abamectin, emamectin benzoate, spinosad, and spinetoram. Blasticidin-S, kusagamycin, streptomycin, oxytetracycline, validamycin, and polyoxins are marketed to control various bacterial and fungal pathogens with stable efficacy [161].

Among the commercialized microbial biopesticides with live cells are the products Serifel^®^, based on *B*. *amyloliquefaciens* MBI600, and Serenade^®^, formulated with *B. subtilis* QST 713. These products control plant diseases caused by bacterial and fungal pathogens. Their ability to colonize plant surfaces through biofilm formation is one of the factors cited by the manufacturers for their activity. Another is that they will stimulate the plant’s resistance mechanisms. Indeed, some products are sold with such claims, such as LifeGard from Certis, based on the protective isolate *Bacillus mycoides*, a phyllosphere colonist [162]. However, findings suggest that they may be less effective than chemical pesticides in the field, perhaps due to poor survival of the microorganism after application, a factor that would lower the production of the bioactive metabolites [163,164,165]. The discussions below indicate the steps between the initial promising laboratory findings and successful commercialization.

As illustrated in Section 2 and Section 3, there is an abundant array of metabolites connected to the secondary metabolism of biocontrol isolates that could be exploited to control multiple targets through ISR or direct antagonism. Dose-dependency is shown in in vitro and *in planta* studies and with different pathosystems. Genome sequencing of increasing numbers of bacteria with soil habitats should promote the prediction of additional active molecules and identify the microbes that produce them. Determining LC/LD50 and MIC values using in vitro assays identifies the metabolites with strong direct effects on targets at low concentrations; these metabolites should be the most lucrative in commercial applications. The bioactive secondary metabolites may have a long shelf life in formulations and persistence in the environment. They can be applied as stand-alone products or to boost the effects of preparations containing appropriate bacterial biocontrol strains.

Maximizing the production of bioactive metabolites can involve different strategies. One approach is adjusting culture methods to optimize metabolite production. A second approach is engineering the genome to overproduce the product or express the biosynthetic genes under a defined productive promoter in an organism for which industrial scale growth is established. Using “smart” promoters with beneficial genes could adjust their production in inoculated plants to only occur under pest challenging conditions. If a pure compound is desired, methods to extract and purify the targeted molecule need to be developed. Shelf life and methods of delivery also should be considered during commercial formulation.

Basic studies reveal that the metabolites are often not constitutively released as cultures grow but are synthesized at specific development times or after a defined environmental stimulus. For example, the *fit* genes are not expressed until the bacterium is within the insect host, and BT toxin synthesis is withheld until the start of spore formation in the *Bacillus* producers [53]. With *Pseudomonas* and *Serratia* isolates, many of the compounds are only produced when quorum sensing is activated, so that there is both a cell density and nutrient dependency. Such understanding comes from basic science studies to establish the knowledge of the lifestyle of the microbe and nutritional effects.

The development of superior culture media is vital. For *Pseudomonas chlororaphis* O6, glucose represses the synthesis of pyrrolnitrin but promotes phenazine production [42]. The concentration of antifungal metabolites produced by *B*. *subtilis* V26 increases up to 93% using potato extract as a carbon source and soybean powder as a nitrogen source. The optimized formulations result in better control of gray mold in tomatoes [166]. The addition of ornithine increases the production of the lipopeptide surfactin by *B*. *amyloliquefaciens* HM618, an effective antagonist for the pathogenic fungi *B*. *cinerea* and *R*. *solani* [167]. The efficacy of *P*. *elgii* HOA73 in controlling gray mold on tomatoes is enhanced by adding chitin during cultivation [168], presumably because it triggers chitinase production. Polyamine additions to cultures of *P*. *chlororaphis* O6 promote phenazine and pyrrolnitrin levels, leading to greater protection against the plant fungal pathogens *R*. *solani* and *Didymella bryoniae* [169].

The media and any stimulants should be cost-effective. Ghribi et al. [170] find that *B*. *subtilis* SPB1 produce more than twice as many lipopeptides through easily available and cost-effective carbon, nitrogen, and inorganic nutrient sources; the increased lipopeptide levels correlate with better control of *Prays oleae* larvae [170]. Waste product use improves cost-efficiency; for example, potato peel and fish wastes enable the production of active lipopeptides with MICs for *Mucor* sp. at 6.25 ppm and *Aspergillus niger* at 12.5 ppm [171]. An extended culture time enhances yield [18] and the product’s potential to thwart pathogenic bacteria and fungi upon application [172].

Programs are under development to systematically modify media to optimize outcomes. The use of “One Factor at a Time” and “Two Factor at a Time” methods by Meena et al. [171] increased lipopeptide output by *B*. *subtilis* KLP2015 by more than four times. Ghribi et al. [170] approach optimization statistically, with suggestions that variable factors such as temperature, moisture, and inoculum age are important. Predicting optimal culture conditions using a statistical approach with the Plackett–Burman design (PBD) increased the production of chitinase and beta 1,4-endoglucanase by *P*. *elgii* PB1 and also enhanced antibacterial activity [173]. The use of PBD shows production of the insecticide avermectin B1 from *Streptomyces avermitilis* 14-12 to be positively influenced by carbon and nitrogen sources [174]. Media composition for producing avermectin B1 has also been optimized using another statistical tool, the Response Surface Method (RSM) [174]. Analysis with RSM additionally optimized the culture parameters for the production of the antimicrobial actinomycin D from *Streptomyces hydrogenans* IB310, an outcome also accompanied by lowered production costs [175].

Several studies have already demonstrated the benefit of genetic engineering in enhancing bioactive products. An early study with *P. fluorescens* CHA0 found that mutations in the gene *rpoD*, encoding the housekeeping sigma factor, enhances the production of the antifungals pyoluteorin, and 2,4-DAPG [35]. The genetic modification of ribosomal structure in *P*. *protegens* Pf-5 is another method for increasing 2,4-DAPG and pyoluteorin levels [176]. Introduction and expression of a *cry* gene enables the synthesis of a BT toxin in a transformed wild type *B*. *velezensis* strain to deliver its native strong antifungal activity and insecticidal efficacy, as demonstrated by its effects on third instar larvae of the diamondback moth *P*. *xylostella* [177]. The transformation of *Pseudomonas synxantha* 2–79, a strong producer of PCA, with genes enabling the production of pyrrolnitrin, extends the protection range of this biocontrol agent to more pathogens in cereals [178]. To heighten its antifungal potential, a combination of ten mutations in regulatory and synthesis pathway genes promotes 2-hydroxyphenazine synthesis in *P. chlororaphis* GP72 four-fold [179]. Changing glycerol utilization in this same microbe improves PCA production [160]. This effort is justified because of the success of the PCA-based pesticide “Shenqinmycin” and the availability of glycerol as a waste product from diesel formation [160]. Other groups have modified different *P. chlororaphis* isolates to produce PCN and 1-hydroxyphenazines in order to enhance the range of plant fungal pathogens that are targeted by one isolate [180,181].

Intact whole live or dead cells could also be important in formulations. Dead cells should still stimulate innate plant resistance mechanisms because the nonviable cells retain structures that act as microbe-associated molecular patterns (MAMPs) [135,182]. Live cells are essential for benefitting the biofilm formation as a root protectant layer and providing continuous sources of VOCs or other biocontrol-active metabolites. Basic research could show mixtures of beneficial organisms that are additive or even synergistic in providing crop protection. A mixture of *Bacillus* species that provides BT toxins and lipopeptides would be a potent weapon for insect and fungal control. Indeed, combined applications of *B*. *pumilus* PTB180 and *B*. *subtilis* PTB185 are compatible with insecticidal activity on aphids in the laboratory and greenhouse [16]. Additionally, a tripartite mixture of an arbuscular mycorrhizal fungus, *Bacillus pumilus*, and *Pseudomonas alcaligenes* enhances the antifungal properties and reduces the root disease caused by the root-knot nematode and a root rotting fungus in chickpea [183].

The studies discussed in Section 2, Section 3 and Section 4 illustrate how partnerships between basic research science on biocontrol microbes and industry can lead to products for agricultural applications that improve plant health. Exchanges of successful methods to produce commercial quantities of reliable cultures of pure or mixed microbes already achieved by the pharmaceutical industry and the agricultural groups are appropriate. For example, probiotics and prebiotics for human and animal health are readily available as “on the shelf” products to purchase.

## 5. Conclusions

It is clear that plants have engineered their own protection schemes by exploiting the intricacies of microbial activities from beneficial microbes that they nurture, especially in the rhizosphere. Emphasized in this review is the fact that rhizospheric biocontrol microbes have the weaponry to be pathogens themselves on multiple plant pests, i.e., fungal pathogens, insects, and nematodes, while having adapted to cohabit with the plant roots such that plant health is promoted. Development is needed to commercialize the varied direct and indirect mechanisms for biocontrol that is revealed by basic research. The current realization of the importance of nurturing soil health for regenerative agriculture aligns well with the use of biopesticides. Consumers are aware of the importance of probiotics for their health, and marketing could better convey their existence and value for plant health in agriculture. Furthermore, the microbes that can control multiple plant pests have a high potential to boost crop output regarding quality and yield at this time when global food security is a major issue.

## Figures and Tables

**Figure 1 microorganisms-10-01053-f001:**
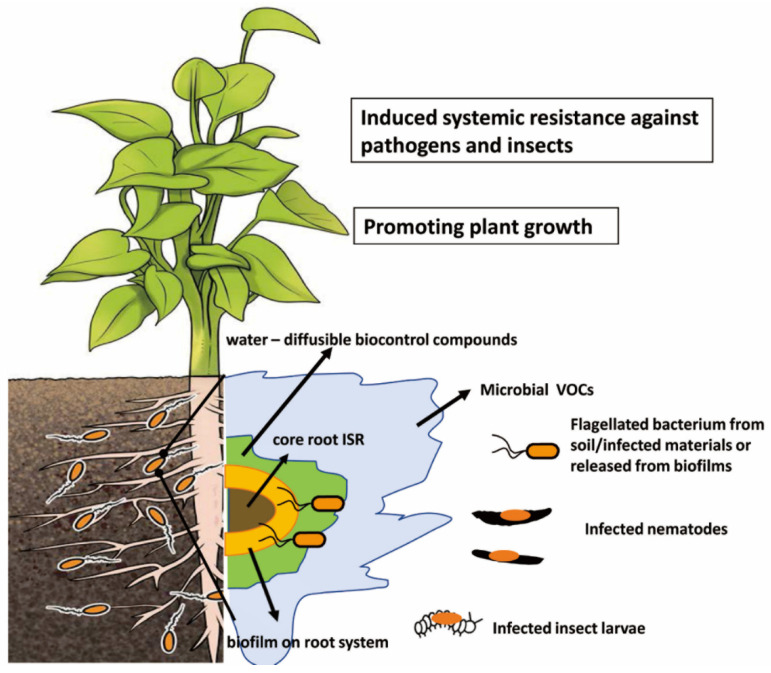
The multiple impacts of biocontrol-active bacteria in the rhizosphere and extending out into bulk soil. Biocontrol microbes are accepted as plant root colonists while exerting antagonism against plant pests through several layers of control in the rhizosphere space. Defense by induced systemic resistance (ISR) may be triggered by metabolites from the biocontrol root colonists. The patchy biofilms on the root surface provide protective barriers. Aqueous diffusion in soil pore waters distributes antagonistic metabolites and enzymes secreted by the biocontrol colonists further into the rhizosphere and biocontrol-active volatiles will diffuse to bulk soil, even to the airspace. These mechanisms boost the survival of the biocontrol bacteria, with their protection within biofilms and spread through chemotaxis and swarming at the rhizoplane. Movement of infected nematodes and insect larvae transport inocula away from the root, and these structures act as hot spots for production of antagonistic products from the biocontrol microbes.

**Figure 2 microorganisms-10-01053-f002:**
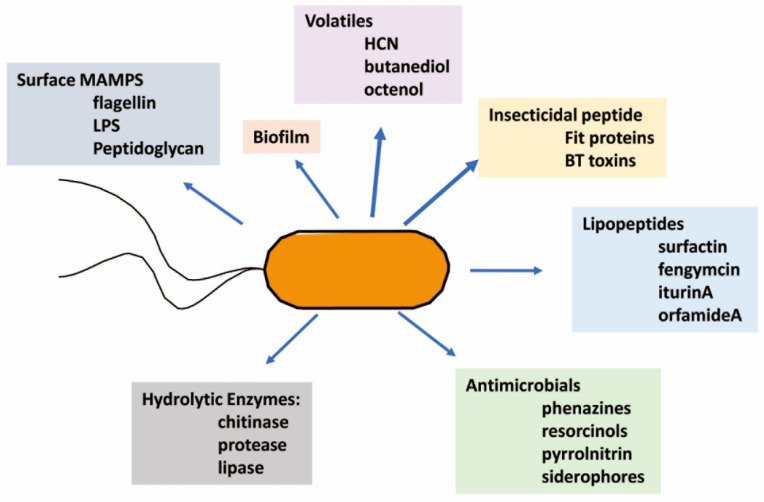
Classes of metabolites involved in biocontrol of microbial, insect, and nematode pests for plants. The biocontrol-active rhizobacteria are highly selective pathogens causing death, or impaired growth, of plant pests through the production of soluble and volatile metabolites, enzymes, and the formation of biofilms. Additionally, microbially associated molecular patterns (MAMPs) and certain metabolites will trigger plant defense systems.

**Table 2 microorganisms-10-01053-t002:** Minimum inhibitory concentration (MIC) and lethal dose (LD)/lethal concentration (LC) 50 of secondary metabolites from microbials against fungal pathogens; lethal dose or concentration against pathogenic insects or nematodes.

Active Metabolite	Pathogen/Pest	MIC (ppm)	LD/LC 50 (ppm)	Inhibition	Source of Active Metabolite	Reference
Lipopeptide	*Tuta absoluta*		180 ng/cm^2^	1st Instar larvae mortality	*Bacillus amyloliquefaciens* AG1	[12]
Lipopeptide	*Fusarium oxysporum* f. sp. *lycopersici*	10		Mycelial growth	*Bacillus amyloliquefaciens* BO5A	[81]
Lipopeptide	*Myzus persicae*		22.2	2nd Instar nymph mortality	*Bacillus subtilis* Y9	[82]
Lipopeptide	*Ectomyelois ceratoniae*		152	3rd Instar larvae morality	*Bacillus subtilis* SPB1	[18]
Lipopeptide	*Spodoptera littoralis*		251 ng/cm^2^	1st Instar larvae mortality	*Bacillus subtilis* SPB1	[17]
Lipopeptide	*Fusarium solani*	3000		Mycelial growth	*Bacillus subtilis* SPB1	[19,20]
	*Rhizoctonia bataticola*	40				
	*Rhizoctonia solani*	4000				
Lipopeptide	*Colletotrichum acutatum*	32		Mycelial growth	*Bacillus subtilis* EA-CB0015	[78]
Lipopeptide	*Colletotrichum* *gloeosporioides*	36.47		Mycelial growth	*Bacillus amyloliquefaciens* MG3	[79]
Lipopeptide	*Phoma tracheiphila*	47.5		Mycelial growth	*Bacillus methyltrophicus* TEB1	[80]
Phenazine-1-carboxylic acid (PCA)	*Botrytis cinerea*	25		Mycelial growth	*Pseudomonas aeruginosa* LV	[83]
PCA	*Botrytis cinerea*	50				
	*Colletotrichum orbiculare*	5		Mycelial growth	*Pseudomonas aeruginosa* GC-B26	[84]
	*Phytophthora capsici*	5				
	*Pythium ultimum*	5				
PCA	*Sclerotium rolfsii*	29		Mycelial growth	*Pseudomonas aeruginosa*	[85]
	*Fusarium oxysporum*	40				
	*Colletotrichum falcatum*	50				
PCA	*Fusarium oxysporum*	1.56		Mycelial growth	*Burkholderia* sp.HQB-1	[86]
	*Colletotrichum gloeosporioides*	6.13				
	*Botrytis cinerea*	1.56				
	*Curvularia fallax*	3.13				
PCA	*Gaeumannomyces graminis* var. *tritici*	1		Mycelial growth	*Pseudomonas fluorescens* 2–79	[87]
	*Rhizoctonia solani*	1				
	*Cochliobolus sativus*	1–3				
	*Pythium aristosporum*	1				
	*Pythium ultimum*	25–30				
	*Pythium uttimum* var. *sporangiifurum*	80–100				
	*Fusarium* sp.	25–30				
Phenazine-1-carboxamide (PCN)	*Botrytis cinerea*	108.12 *		Mycelial growth	*Pseudomonas aeruginosa*	[88]
PCN	*Rhizoctonia solani*	5		Mycelial growth	*Pseudomonas aeruginosa* MML2212	[89]
1-Octen-3-ol	*Tribolium castaneum*		16.75	Adult mortality	*Paenibacillus polymyxa* BMP-11	[30]
Benzothiazole	*Tribolium castaneum*		3.5	Adult mortality	*Paenibacillus polymyxa* BMP-11	[30]
2,4-Diacetylphloroglucinol (2,4-DAPG)	*Fusarium oxysporum* f. sp. *lycopersici*	16		Mycelial growth	*Pseudomonas fluorescens* CHA0	[90]
	*Gaeumannomyces graminis* var. *tritici*	16–32				
	*Pythium ultimum*	64				
	*Rhizoctonia solani*	32–64				
2,4-DAPG	*Xiphinema americanum*		8.3	Adult mortality	*Pseudomonas fluorescens*	[91]

* EC50 (ppm).

## Data Availability

Not applicable.

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
