# Peer review of "Root-Associated Bacteria Are Biocontrol Agents for Multiple Plant Pests"

_microorganisms, 2022, doi:10.3390/microorganisms10051053_

Round 1

Reviewer 1 Report

The manuscript “Root-associated bacteria are biocontrol agents for multiple plant pests” reviewed microbes that could potentially be formulated into pesticides against various microbial plant pathogens, insects and nematodes. Since similar reviews have been published several times, such as the use of PGPR, biocontrol agents and etc., the manuscript should pay attention to the research progress in Microorganisms as Biocontrol Agents in the recent five years. I suggest the manuscript be revised and the authors adjust the content accordingly.

Some points of concern:

  1. The topic of the manuscript is about root-associated bacteria and biocontrol agents, so it is not suggested to discuss the secondary metabolites or agricultural antibiotics. There are too many varieties of secondary metabolites or agricultural antibiotics, and they could be reviewed specifically.
  2. In Table 1, It’s P. chlororaphis PCL1391, not CL1391; Line 452, not P. chlororaphis PCL.
  3. In Table 2, some metabolites are same, and the pathogens are same, too. Why are their MICs different?
  4. Line 84, the information in this review suggests that the biocontrol agents are pathogens. What does it mean?
  5. Line 360-361, Shenqinmycin is chemically synthesized. But I know Shenqinmycin is actually produced by fermentation.
  6. Line 547, the format is wrong.
  7. Some sentences in the manuscript are not easy to understand, for example, Line 692-694. It is suggested to check again.

Author Response

  1. The topic of the manuscript is about root-associated bacteria and biocontrol agents, so it is not suggested to discuss the secondary metabolites or agricultural antibiotics. There are too many varieties of secondary metabolites or agricultural antibiotics, and they could be reviewed specifically.

Answer: The intention of the review was to address biocontrol in the rhizosphere and the fact that control of multiple plant pests by single rhizosphere colonizing bacteria is found.  We synthesized information from specific research findings to help the reader realize three main points:

  • biological control does involve “the secondary metabolites “ produced by biocontrol agents
  • many of these secondary metabolites act not only to control microbial plant pathogens but also insect and nematode plant pests
  • the effective doses of the active metabolites are often similar to those of commercial products.

The goal of the paper was not just to list the secondary products but to emphasize the breadth of rhizosphere activities which connect biocontrol agents with plant health. And we focused on metabolites that control both insect, nematode and microbial plant pests. This concept has developed from the breadth of studies on biocontrol organisms.  

As noted by Reviewer 1, there are many such reviews already where the secondary compounds are discussed by structure, by source and their pest targets. Purposefully our review was approached by consideration of the microbial genera that are biocontrol agents rather than the secondary compounds. We illustrate that many of the biocontrol active metabolites are produced by more than one genus of biocontrol bacteria. Then we expanded the review to discussion of how metabolites and/or the biocontrol agents can be commercialized. We justify the importance of this discussion because we have to connect the academic findings to commercial field applications to address the food needs of the expanding global population. In this light we also have discussed the importance of biopesticides in current and future agriculture in part because current chemical pesticide control mechanisms are failing soil and plant health as well as raising consumer concerns.

Generally reviews reflect what is important to the authors. The value of special topic editions that focus on a specific topic is that the issues expose readers to different viewpoints. Review papers are not confined only to a very specific set of data, this is the objective of the research papers. We realize as expressed by the reviewers that not all will have the same response to a special topic call.

  1. In Table 1, It’s P. chlororaphis PCL1391, not CL1391; Line 452, not P. chlororaphis PCL.

Answer: Corrected throughout manuscript.

  1. In Table 2, some metabolites are same, and the pathogens are same, too. Why are their MICs different?

Answer: Corrected. The MICs differ due to there being no standardization in assessment techniques. In the review we document how the growth medium and time of culture will influence the biocontrol potential of a microbe. The extent of purification will also change effectiveness of a metabolite because it is seen that combinations of metabolites act synergistically to improve control over a single constituent. Additionally, the target also is the cause of variability with their being differences in isolates (e.g., genomes of bacterial pathogens or fungal pathogens of the same genus and species can have large differences in gene numbers and functions) as well as in the methods to raise and test the antimicrobial effects. Biocontrol also is influenced by the host plant when the studies extend to how the plant is protected. 

  1. Line 84, the information in this review suggests that the biocontrol agents are pathogens. What does it mean?

Answer: Pathogenesis is defined as an interaction that results in disease. Consequently, the killing/growth damage of a bacterial or fungal plant pathogen or an insect or a nematode by another microbe is an act of pathogenicity. The biocontrol agents have a defined set of hosts for which they have pathogenic potential. We think of them as being nonpathogens to plants and this explains the thought that they are not pathogens. Evolution has enabled the biocontrol agents to have minimal negative consequences to plants but strong negative impact on plant pests.

  1. Line 360-361, Shenqinmycin is chemically synthesized. But I know Shenqinmycin is actually produced by fermentation.

Answer: We greatly appreciate this information which does fit with the concept that glycerol is an important raw material in its production. The sentence is changed to “The commercial product Shenqinmycin contains PCA as the major metabolite and has high efficacy in the field against Phoma infections [105]”.

  1. Line 547, the format is wrong.

Answer: The type set is corrected to Palomino type 10

  1. Some sentences in the manuscript are not easy to understand, for example, Line 692-694. It is suggested to check again.

Answer: We thank the reviewer for pointing out problems in communication and have made corrections in many sentences including the lines above. “The studies discussed in Sections 2 through 4 illustrate how partnerships between basic research science on biocontrol microbes and industry can lead to products for agricultural applications that improve plant health.”

We also have amended the document to clarify sentence construction etc and these changes are noted in returned text.

Reviewer 2 Report

Generally, a well-written article on the important topic of microorganisms and, consequently, their metabolites that can be used as biological plant protection products. They are active against various microbial plant pathogens and destroy insects and nematodes. They colonize the rhizosphere where they promote healthy plant growth through various mechanisms. They are pathogenic to plant pathogens, including microbes, insects and nematodes. The metabolites released from useful strains inhibit the growth of bacterial and fungal plant pathogens and participate in the toxicity of insects and nematodes. This work discusses a wide range of metabolites involved in plant protection. The focus was on firmicutes and pseudomonads. The review assesses how cultivation conditions can be optimized to provide formulations containing preformed active metabolites for rapid control, with or without viable microbial cells as inoculum, to increase plant productivity under environmental field stress conditions. Metabolites produced by microorganisms can be the equivalent of probiotics and prebiotics for humans and animals.

Minor comments:

Line 547 – Bacillus subtilis Bacillus amyloliquefeciens – are written in different font

Author Response

Concerns

Line 547 – Bacillus subtilis Bacillus amyloliquefeciens – are written in different font

Answer: Corrected and also see Reviewer 1 reply.  

Reviewer 3 Report

This review focuses on microorganisms (bacteria) that act as biocontrol agents against insects, nematodes and fungal pathogens.  The authors generalise about the genera having an ability to perform biocontrol activities - this ability is isolate specific.  The majority of examples used are laboratory based - I would have liked to have seen a section where the examples cited have been used in field applications (the ultimate goal of biocontrol).  I have the following comments:

Where indicated whether or not the organism is Gram negative or positive - Gram should start with a capital letter has it is the name of a person.

One cannot start a new section with the words "likewise" or "also" rather state "as indicated previously".

Many paragraphs have no references (important in a review) and thus there are statements made that may or maybe supported by literature.

line 15 not all rhizosphere inhabitants are pathogenic

line 17 sentence makes no sense - what do you mean by "feed forward"?

line 24 why environmental stress? surely they function with or without stress?

line 32 induced resistance is the target? sentence makes no sense

line 38 it is isn't clear whether or not fluopyram is a metabolite from a biocontrol agent...

Line 50 what do you mean by agricultural site?

line 52 I disagree - even without climate change they would be influenced by the environment

line 55 and 56 link between sentences? random insertion of a sentence without linkage to the former or latter sentence

line 72 not genera but species

what do you mean by the term "probiotic"? used more than once in the review

line 84 sentence as it stands makes no sense - rather link it to the next sentence

Legends for both figures 1 and 2 are too long and the text should be included in the body of the review (with references).

line 201 an example of a sentence requiring a reference

line 207 what do you mean "a boon"?

spp. should not be in italics

line 274 does it target F. oxysporum spores and not mycelium? sentence is not clear especially in light of the sentence that follows

Line 249 "in hoop"?

line 361 fermentERS

line 511 not all psuedomonads and Bacillus spp. induce systemic resistance.  Some pseudomonads are plant pathogens

Author Response

This review focuses on microorganisms (bacteria) that act as biocontrol agents against insects, nematodes and fungal pathogens.  The authors generalise about the genera having an ability to perform biocontrol activities - this ability is isolate specific.  The majority of examples used are laboratory based - I would have liked to have seen a section where the examples cited have been used in field applications (the ultimate goal of biocontrol).

Answer: We agree that there is a gap between lab and pot studies and field applications and we tried to point out that problem. As noted in paper there are commercially available biologicals, or example Bt applications are widely accepted as are whole cell products for cereal disease. There are fewer examples of metabolite field use – outside of Shenqinmycin (PCA-based)- in part because it is costly and almost impossible from the needs of scale up (flask culture to fermenters) for a research lab to make enough of the metabolite to use on a field scale.

I have the following comments:

Where indicated whether or not the organism is Gram negative or positive - Gram should start with a capital letter has it is the name of a person.

Answer: All corrected in text as highlighted

One cannot start a new section with the words "likewise" or "also" rather state "as indicated previously".

Answer: We thank you for this improvement to the flow of the sentences. The two sentences beginning with likewise are modified as seen at lines 454 and 548.

Many paragraphs have no references (important in a review) and thus there are statements made that may or maybe supported by literature.

line 15 not all rhizosphere inhabitants are pathogenic   

Answer: Agreed. We did not intend for this interpretation. Sentences at lines 14 and 21 have been modified.

There is a general belief in the statement that biological control organisms are not pathogenic but that is not true. They are not pathogenic on plants but they do damage to other organisms – the direct biocontrol activity requires that to happen. Ie they have a set of pathogenicity genes. These make the active secondary products and other factors that control growth and function of their targets.

line 17 sentence makes no sense - what do you mean by "feed forward"?

Answer: Sentence has been modified. The use of the term was to emphasize that the plant supports cohabitation of these biological control agents with no disease symptoms in the plant.  Some of these microbes (as discussed) turn around with “feed forward” promotion of plant health by triggering the plants own immune systems (ISR etc). Thus, the act of feeding the microbe with C and N from the root exudates leads both directly and indirectly to better plant health. This protected plant will be better able to support the populations of the beneficial microbe than a plant that dies through disease. This is a “feed forward” system. But we have removed the term feed forward in the reconstructed sentence.

line 24 why environmental stress? surely they function with or without stress?

Answer: Concluding sentence has been modified removing the term environmental stress. But not sure about this—if the plant has optimal growth conditions would you see an effect of biocontrol agents? By definition of the term biocontrol it is an isolate that improves plant by controlling a plant pathogen. In early 1970 discussions of biocontrol (at that time only direct effects were known) it was thought that improved plant growth was due to the biocontrol agents suppressing uncharacterized low levels of plant pathogens. This possibility still could exist. However, improved growth effects can sometimes be due to the PGPR contributing to better nutrition (e.g., release of Fe or P from soil minerals). Dangl’s group has shown connection with P deficits and root colonization by organisms that help provide P. In agriculture it is rare that the growth conditions (from weather, water quality and amount to soil health) are optimal and the root colonists help to buffer problems from such “environmental” stress.  

 line 32 induced resistance is the target? sentence makes no sense

Answer: Sentence modified “development of resistance to the pesticide in the targets”

line 38 it is isn't clear whether or not fluopyram is a metabolite from a biocontrol agent...

Answer: The sentence wording changed to show it is not a metabolite from biocontrol agent. One recent exception is the synthetic pesticide fluopyram which is an example of dual efficacy to control both fungi and nematodes as a fungicide and a nematicide.

Line 50 what do you mean by agricultural site?

Line 52 I disagree - even without climate change they would be influenced by the environment

Answer: Yes we agree that problems exist other than those caused by climate change. Wording of sentences in lines 50-55 has been changed.

Line 50 “However, application of biopesticides into the agricultural fields has drawbacks that research involved in formulation, storage, and applications need to overcome. In the field, the survival and efficacy of microbial inocula may be limited by environmental factors. References were added. For example, agricultural chemicals, may negatively influence the activity and survival of beneficial metabolites in the field.”

line 55 and 56 link between sentences? random insertion of a sentence without linkage to the former or latter sentence

Answer: Section redrafted to explain the comment that appeared random. ie lines 57 to 61. “Compared with the rapid killing effects of some pesticides, time is required to amass beneficial microbial communities in the rhizosphere. The early observations of biological control of the take-all fungus, Gaeumannomyces graminis, on cereals demonstrate how time and repetitive cropping are required to establish a suppressive soil effective for this pathogen [6].”

line 72 not genera but species

Answer: Section 2 is broken up first by genera but then includes examples of activities from different species of these genera. Sentence line 72 now reads “Section 2 of the review introduces major bacterial genera and species for which biocontrol”

what do you mean by the term "probiotic"? used more than once in the review

Answer: The premise that plant health like health in mammals etc is due in part to their interaction with microbes termed probiotics is present in the plant literature. The term is used in papers from 2012. It is a pertinent term – it would encompass plant growth promoters and agents that deter pathogen growth. A definition from NIH is “Probiotics are live microorganisms that are intended to have health benefits when consumed or applied to the body.” So by comparison the biological control microbes act as plant probiotics.

We have added references that include a book publication where they specifically use the term probiotics

Added references are:

  1. Berlec, A. Novel techniques and findings in the study of plant microbiota: Search for plant probiotics. Plant Sci. 2012. 193–194, 96-102.
  2. Spence, C.; Alff, E.; Shantharaj, D.; Bais, H. Probiotics for plants: Importance of rhizobacteria on aboveground fitness in plants. In Bacteria in Agrobiology: Plant Probiotics. 2012,1–14, Springer publisher.
  3. Woo, S. L.; Pepe, O. Microbial consortia: Promising probiotics as plant biostimulants for sustainable agriculture Front. Plant Sci. 2018, 9, 1801.

In addition, we thank the reviewer for their comments at this point in the paper and the section beginning at line 80 has been rewritten for clarification.

line 84 sentence as it stands makes no sense - rather link it to the next sentence

Legends for both figures 1 and 2 are too long and the text should be included in the body of the review (with references).

Answer: Legends are shortened and they are based on text already in place with references. The understanding that we have as authors and how we have always addressed figure legends is that the legend should be complete enough for understanding without reading the text.

line 201 an example of a sentence requiring a reference/ line 207 what is meant by a boon

Answer: Text modified and references to the nutritional value of insects provided.

  1. Oonincx, D. G. A. B.; Finke, M. D. Nutritional value of insects and ways to manipulate their composition. J. Insects Food Feed, 2020, 7, 639-659
  2. Bukkens, S. G. F. The nutritional value of edible insects,Ecol. Food Nutr. 1997. 36, 287-319.

line 207 what do you mean "a boon"?   

Answer: From the Oxford dictionary, boon is a noun and a thing that is helpful or beneficial. "the navigation system will be a boon to both civilian and military users"

line 274 does it target F. oxysporum spores and not mycelium? sentence is not clear especially in light of the sentence that follows

Answer: Section has been modified and the lipopeptides from Bacillus were tested for conidial spore germination.

Line 249 "in hoop"?

Answer: Hoop referred to hoop houses  where bent tubing covered with fabric are used to raise protected crops -  because of cost these hoop houses  are becoming more used than glass-framed green houses. However, the term hoop has been omitted. It is a common term in growers use. Tunnel houses are another name.

line 361 fermentERS

Answer: Corrected by changes in wording. Ferment was used as a noun

line 511 not all psuedomonads and Bacillus spp. induce systemic resistance.  Some pseudomonads are plant pathogens

Answer: We agree with your comment, especially that there are plant-pathogenic pseudomonads.  Our statement was to say that certain antifungal metabolites will trigger induced systemic resistance. We have added the word certain and added a sentence change.

Reviewer 4 Report

The authors of this manuscript present an interesting review on Root-associated bacteria biocontrol agents for multiple plant pests. Introduction and the rest of the sections are well described. Also, presented tables are clear and quite explicable. I believe that this review can add further research interest in root-associated bacteria biocontrol agents for multiple plant pests.

Abstract

COMMENT:

The abstract describes sufficiently the findings of this review.

Introduction

Introduction section is well written and, in my opinion, give the appropriate information without being extended. The purpose of the research work is clearly presented.

Line 40       Flyopyram. Is it allowed in E.U?

The array of isolates and their products for biocontrol of multiple targets  

Table 2     ng/cm2

Figure 1    I think that the title of figure 1 is too extended.

Induction of plant systemic resistance mechanism by biocontrol agents

LINE 547….please check text style

Conclusions

The conclusion section described sufficient the findings of the review

References

COMMENT:

Although I couldn’t notice any mistake in reference style, I suggest to check reference list once again.

In spite of the manuscript is very clear and carefully written some improvement it should be done.

Author Response

Line 40  Flyopyram. Is it allowed in E.U?

Answer: From web sources Fluopyram is accepted in the EU although it seems that modifications to its use are constantly being added. One recent entry is as follows. Review of the existing maximum residue levels for fluopyram according to Article 12 of Regulation (EC) No 396/2005 Published: 14 April 2020 Approved: 6 March 2020  EFSA Journal 2020;18(4):6059. DOI:  https://doi.org/10.2903/j.efsa.2020.6059

 The array of isolates and their products for biocontrol of multiple targets  Table 2     ng/cm2

Answer: Corrected by use of superscript ie cm2

 Figure 1    I think that the title of figure 1 is too extended.

Answer: The title is now less than one line and the description has been shortened

LINE 547….please check text style

Answer: Corrected with font change.

Round 2

Reviewer 1 Report

The authors have revised the manuscript and improved the quality of manuscript, responded to the concerns proposed by the reviewer and given their explanations. The reviewer also commented that since similar reviews have been published several times, such as the use of PGPR, biocontrol agents and etc., the manuscript should pay attention to the research progress in Microorganisms as Biocontrol Agents in the recent years. But the authors did not reply and made no revision about this viewpoint. I suggest the manuscript be revised again.

About the explanation made by authors, there are two questions to discuss.

1. About the literature review to be published in one specific journal, due to the space limitation and readers in related research, maybe it is better to write a concise overview or a mini review.

2, the authors thought that the biocontrol agents have a defined set of hosts for which they have pathogenic potential. Yes, I agree. But in the manuscript, the word pathogen only represents plant pathogen, otherwise there will be misunderstandings between beneficial microbes and plant pathogens.

Author Response

The authors have revised the manuscript and improved the quality of manuscript, responded to the concerns proposed by the reviewer and given their explanations. The reviewer also commented that since similar reviews have been published several times, such as the use of PGPR, biocontrol agents and etc., the manuscript should pay attention to the research progress in Microorganisms as Biocontrol Agents in the recent years. But the authors did not reply and made no revision about this viewpoint. I suggest the manuscript be revised again.

About the explanation made by authors, there are two questions to discuss.

Point 1;  About the literature review to be published in one specific journal, due to the space limitation and readers in related research, maybe it is better to write a concise overview or a mini review.

Response 1; We were not asked to write a mini review, but a review. Our concept was to convey to the reader the following points that offer novelty to this paper:

    1) The many examples of biocontrol of multiple types of plant pathogens, across kingdoms, that are achieved by a single biocontrol bacterium from the rhizosphere. Often in the original research papers the biocontrol covers only one designated pathogen and frequently pertinent findings occur in specialized journals. Thus, it is easy for a reader who follows fungal plant pathogens not to be aware of what is happening to insect larvae when exposed to the same metabolic pool from the biocontrol agent.

2) A summary showing how the active metabolites / mechanisms of control fall into similar classes for the different microbes; here we discussed mode of action as well, for example showing those that attack membranes versus causing ROS stress.

   3) Similarity in the potency of isolated metabolites with antagonistic potential to commercial pesticides. This point is often overlooked in the research papers; many of these papers do not include commercial pesticides for comparison. Indeed at growers meetings comment like the following are heard- “those biological controls just do not work” , “they lack the strength of a commercial fungicide.” We provide many examples of where the metabolites are comparable even better.

  4) The gap area between the extensive scientific knowledge and commercialization of biopesticides. We wished to present a review that extended the discussions to be more than just academic factual summary. To this end in this section we linked information to basic science of the biocontrol agents i.e., what genes concerned with synthesis can be genetically engineered? Or what is the optimal medium for growth?

    As authors we were not advised by the journal of other papers that are being considered for publication in this special edition. We had no indication of how our paper would align with others, either in terms of its complementarity or in overlap of material; this is quite different from book chapter submissions where your discussion angle is often quite delineated. However, two of us are educators and we believe that learning is advanced from seeing and integrating different viewpoints. A direct inquiry to an editor of MDPI reviews stated there is no page limit for reviews; our paper is 25 pages. A survey of five papers accepted in Microorganisms as Reviews show a range of 16 to 33 pages in length. Indeed, a review for the same edition (Jamil et al.,) is 26 pages in length.

Point 2, the authors thought that the biocontrol agents have a defined set of hosts for which they have pathogenic potential. Yes, I agree. But in the manuscript, the word pathogen only represents plant pathogen, otherwise there will be misunderstandings between beneficial microbes and plant pathogens.

Response 2; We clarified the word plant pathogen and beneficial biocontrol agents as pathogen to insects and nematode throughout the manuscript.

Reviewer 3 Report

I believe the authors have addressed my concerns satisfactorily.

Author Response

Point 1; I believe the authors have addressed my concerns satisfactorily.

Response 1; Thank you very much for your kindness and your efforts to review on submitted manuscript. Reviewer 3 had no further comments because we had addressed their concerns well in this revision 1.